# Reprocess: Process Refinement for Improving Accuracy in Hybrid Planning Domain Models

**Alan Lindsay[1], Santiago Franco[2], Rubiya Reba[1], Thomas L. McCluskey[1]**

[1]University of Huddersfield, UK.
`a.lindsay@hud.ac.uk`
[2]Royal Holloway, University of London, UK.
`santiago.francoaixela@rhul.ac.uk`

## Abstract

The creation and maintenance of a domain model is a well recognised bottleneck in the use of automated planning; indeed, ensuring a planning engine is fed with an accurate model of an application is essential in order that generated plans are effective. Engineering domain models using a hybrid representation is particularly challenging as it requires accurately describing continuous processes, which can have complex numeric effects. In this work we consider the problem of the refinement of an engineered hybrid domain model, to more accurately capture the effect of the underlying processes. Our approach exploits the information content of the original model, utilising machine learning techniques to identify important situation and temporal features that indicate a variation in the original effect. We use the problem of modelling traffic flows in an Urban Traffic Management setting as a case study and demonstrate in our evaluation that the refined domain models provide more accurate simulation, which can lead to higher quality plans.

## Introduction

The modelling problem is a well recognised bottleneck in the use of automated planning: if the problem formulation (domain model, initial state, goal) does not correspond to the problem at hand then clearly the wrong planning problem will be attempted. For the planner to be able to generate accurate plans, a domain model used by a planner must adequately model the application domain; in particular the domain model, utilising its operational semantics, must simulate the effect of a plan accurately. Various lines of research have focused on the domain model accuracy problem, from as far back as Benson's thesis (Benson 1996) to more recent work on Space applications (Clement et al. 2011; Frank 2015).

Even in classical planning, validating that the model is an accurate representation may be a challenge. If the model is hybrid, involving continuous changes to variables, the burden on the knowledge engineer is greater than in a traditional classical case, since the encoding of a continuous changes would seem to be more difficult than the encoding of discrete changes. This is particularly apparent in applications to physical systems where it can be necessary to use richer languages to capture the important phenomena of an environment. Further, it may be that modelled processes change

their behaviour over time, and the model has to be constantly maintained. Or it may be that there are many similar processes in the domain, but each is a variant over the space that the planning function is aimed at. In these cases, producing a faithful hybrid planning domain model is indeed a challenge.

Our research is aimed towards creating a general method for the refinement of engineered hybrid domain models, suitable for automated planning in real world environments. In this paper we describe a method for the refinement of domain model *process* descriptions, utilising the PDDL+ language for model encoding. The approach is designed so that the resulting refined domain model will retain (a) its original language - PDDL+ (b) its efficiency, so that plan generation and simulation times will not be greatly increased (c) its readability so that changes can be explained in terms of high level features. We illustrate and evaluate the approach using as a case study the domain of Urban Traffic Management. A PDDL+ model of traffic flow is utilised based on earlier work in this area, and an industry standard, proprietary simulator called AIMSUN (Barceló and Casas 2005) is used to provide process training data. Our evaluation demonstrates that simulation using the refined domain model is closer to the behaviour of the actual processes being modelled, and when using expert selected features the plans produced with the refined domain are of higher quality than the original. Further, we show that the efficiency of the model increases linearly with complexity.

## Preliminaries

We are concerned with applications involving both plan generation and plan execution, where in execution mode an oracle is available that can supply the current state of the system. The oracle may obtain state information from a stream of sensor information from the environment, or be supplied by a simulator which represents the real environment (Clement et al. 2011). This is normally required in planning and execution so that the planner can monitor a generated plan's execution, and check it is having the expected effect, with the possibility of re-planning if actual and expected diverge. Additionally, in our approach, this state information is used to drive refinement of the domain model.

The method used in this paper is summarized as follows: a planning application is being undertaken in which knowl-

edge engineers have developed a hybrid domain model $DM$ to the point where an available planning engine can generate plans for the application, but the plan engine's simulation does not accurately match the state information from the oracle due to inaccurate process descriptions. Given a process $P$ in the $DM$ changes a set of continuous variables $V$, for each $v$ in $V$, assume the effect of $P$'s corresponding domain process is dependent on some subset of state variables captured in $DM$. From the environment, training sequences of external states are obtained, each annotated with which processes are in operation at that time. A learning method uses them to discover an expression, incorporating a combination of state variables from $DM$, which best predicts the effects on $v$ as found in the external states, and replaces the original expression with the discovered expression in the process specification. The domain model is then replaced with the new version.

To make this method concrete, and evaluate it on a real planning application, we have used the PDDL+ encoding (Fox and Long 2002) for hybrid domain models, and the ENHSP planner (Scala et al. 2016) to generate plans with the engineered and refined domain models. In a PDDL+ encoding, a hybrid planning model involves time-dependent discrete-continuous changes in the numeric resources which can be encoded with three main components: processes, actions and events. A process simulates continuous changes on numeric variables which can be initiated or terminated by an action/event whereas an event can be triggered by the external environment to bring about discrete changes.

**Motivating Example Domains:** Consider as a simple example a model of the Bouncing Ball domain[1] where the ball has perfect elasticity i.e. no energy is lost upon bouncing. The original process describing the ball movement is called ball-movement, and may be engineered using the known gravitational equation. If it were possible to collect accurate and representative data on the position and velocity of the ball, the model could be refined to better fit reality - in this case we assume by finding the appropriate falling friction coefficient, e.g. 0.8. However, when the ball is rising, both gravity and friction pull the ball back, resulting in a rising friction coefficient bigger than one, e.g. 1.2. In other words, the model would be refined in accordance with the data by learning two scaling factors to the increase in velocity splitting the ball-movement process into two new processes by adding as preconditions whether $velocity >= 0$ or $< 0$.

As another example, consider a fixed angle solar panel used for charging[2], where the original engineered model assumes a constant rate of charge. The amount of electricity it receives, however, is a function of the time of the day, weather etc. Given a set of training data containing time, weather, etc, and the charging effect, the PDDL+ domain model can be refined into multiple charge processes to improve accuracy. This example illustrates a potential trade-off

_____________

[1] Similar to the example in Coles's tutorial http://cognitive-robotics17.csail.mit.edu/docs/tutorials/Tutorial8_Planning_in_Hybrid_Domain.pdf

[2] A good example is the satellite cooled domain, see https://nms.kcl.ac.uk/planning/software/colin.html

between accuracy vs computational complexity: the more accurate, the more processes need to be checked for applicability.

As an in-depth case study, for the rest of this paper, we consider the domain of Urban Traffic Management (UTM). (Vallati et al. 2016) introduced a hybrid planning approach using a PDDL+ representation to solve traffic management problems involving congestion in both unexpected and regular road traffic. In their macroscopic model of UTM, a road network is represented by a directed graph where the edges and vertices denote the road links and junctions (the entry/exit point of the road links) respectively. Each entry/exit pair is called a turn and the traffic flow through a turn (here called the *turn rate*) is measured in standardized vehicles (PCU) per second. Flow across a junction is organised by grouping these turns into *stages* (representing the stages of traffic signals) and then selecting durations for each of these stages. Each road link has a maximum *capacity*, and the *occupancy* of a road link denotes the current number of vehicles within it. This approach was used in feasibility trials and incorporated a simple description of the turn rate process which took no account of the changing features of the road network (McCluskey and Vallati 2017). The case study demonstrates the utility of learning a time/situation dependent turn rate which leads to more accurate simulation and better solution plans.

## Process Refinement for PDDL+

Our approach is to replace a process with a set of more targeted processes that will each better capture the effects of the observed process in a specific situation. One important feature of our approach is that we learn effect modifying factors, which instead of overwriting the original specified effects, adjusts them. The motivation here is that in focusing on refining the process descriptions we exploit the existing structure and information content within the existing model.

The general template for our target representation is presented in Figure 1 for the `flowrun_green` process, which models flow of traffic across a junction as a continuous process. The intention is to form a partitioning that identifies groups of states that require a similar modification to the effect. For example, a process might require a warm up period, which can be better represented by separating the process's first $n$ seconds of operation and reducing the process's effect in that initial period. This is achieved by constructing a set of processes that each define the process's effect for a more specific situation. This requires that the new processes have extended preconditions, as illustrated by the `refinement-condition` predicates in Figure 1. The new process also defines factor terms (highlighted in red), which are specified for each of the continuous effects of the process.

The context that we have selected for process refinement includes both temporal and situational aspects, which can both impact on the effect of continuous transitions.

**Temporal Features:** process effects can change depending on how long the process has been active. E.g., PCUs

getting up to speed when a stage becomes active. In order to allow the refined processes to reflect this change, the context includes the time since the process became active. Notice that this part of the context is treated in the same way as the functions in the state features.

**State Features:** a process's effect may also be different when certain state relationships exist (e.g., reduced flow for turning across a busy road). The context that we use for learning includes all of the not redundant predicates, both numeric and propositional, that can be referenced by the parameters of the process. A predicate is redundant if its value does not vary between different activations of the process (e.g., `active`, will always hold for the process's stage, $p$, while the process is active).

We use only the parameters of the original process for the following reasons. We observe that the factors that impact on a processes' behaviour are most likely related to the parameters of the process. Moreover, by focusing the context it should allow less data to be used, while still learning process specific phenomena. For example, there will be fewer correlations with external features that might be specific to the gathered data. A final reason is that including additional parameters further increases the computational complexity of the model.

**Feature Space** The feature space is constructed automatically from the temporal and state features of the domain. Propositional terms are included directly from the context. The numeric features are built from any function in the context (denoted $f[y_0, \ldots, y_n]$) using the following language:

$$T[X] ::= (T[X]\, R\, T[X]) \mid f[y_0, \ldots, y_n], y_i \in X$$
$$R ::= + \mid - \mid * \mid /$$

We define the depth of a numeric term $depth(T[X])$, as the count of the number of function terms in T[X], e.g., $depth$[(+ (occupancy ?l1) (occupancy ?l2))]= 2. The possible features for each node are then generated by fully expanding the expressible terms in the language up to a defined depth. In this case this constructs expressions that build from the functions of the domain using the typical binary relationships: $\{+, -, *, /\}$.

### Refinement as a Learning Problem

In this work we map the problem to a multi-target regression problem. The aim of these problems is to learn a model that relates the set of input features to a set of target outputs. There is one target output for each of the process's continuous effects. This allows us to learn a single model for each process, which can be more concise than if a model is learned for each target variable, but it is also more practical for encoding back into PDDL+. The learned model can be used to identify the appropriate factor for each effect at any state.

We modify the input data into a process orientated form, for a process $P$. For each time point, $t$, and active process, $p$ (an instantiation of $P$), we have a learning example, $\langle p.X, t_{active}, s \rangle$, which describes, $p.X$: a process header,

```
(:process flowrun_green-leaf-i
    :parameters (?p - stage ?r1 ?r2 - link)
    :precondition
    (and
        (> (occupancy ?r1) 0.0) (active ?p)
        (>= (turnrate ?p ?r1 ?r2) 0.0)
        (< (occupancy ?r2) (capacity ?r2))
        (refinement-condition-1 ?p ?r1 ?r2)
        ...
        (refinement-condition-n ?p ?r1 ?r2))
    :effect
    (and
        (increase (occupancy ?r2)
            (* (#t (* refinement-factor-i-1 (turnrate ?p ?r1 ?r2))) )
        (decrease (occupancy ?r1)
            (* #t (* refinement-factor-i-2 (turnrate ?p ?r1 ?r2))))))))
```

Figure 1: An example of process refinement using the flowrun_green process. A set of conditions (bold) identify sets of states where it is appropriate to modify the process's effect using a factor (red).

$t_{active}$: the active time of the process and $s$: the state. Each of these examples is used to construct a single data point, as described here. The set of features: the set, $T[X]$, expanded to depth, $\delta$ and single predicate terms (as described above), define lifted terms (e.g., $(/(occupancy\,?r1)(capacity\,?r1)))$). The terms' values are calculated for a specific example, $\langle p.X, t_{active}, s \rangle$, by first instantiating the expression using the parameters, $p.X$. The instantiated expression is then evaluated in $s$.

For each data point we also compute a set of targets. A distinct factor is computed for each of the process's effects ($e \in EFF(P)$). For each data point, $d$, the observed effect of the process, $obs_d(e)$, and the modelled effect of the process, $mod_d(e)$ are used to compute the factor, $factor = \frac{obs_d(e)}{mod_d(e)}$.

Finally we prune examples and features as follows: i) We prune any features where any evaluation leads to an undefined value (i.e., the result of the expression is not defined). This would suggest that the feature might lead to undefined values during planning. ii) We prune any examples, where the target is not defined (i.e., where the modelled effect is 0).

The result is a consistent set of features and targets for examples across all of the instantiations of the process that can be addressed as a general machine learning problem.

### Regression Tree Learning

The hypotheses for the process model are represented by regression trees. Regression trees can be used to approximate complex functions and algorithms exist that can grow them efficiently and effectively from observation data. Moreover it is possible to encode the learned model in PDDL+, as presented at the end of this section. This contrasts with several of the alternative representations used for machine learning. In this section we overview our tree learning approach and describe the aspects that are important for our approach.

```
function Reprocess(obsTrn, obsVal, P, DM) :
    F = chooseFeatures(DM)
    trnData = makeProcessOrientated(trnObs, P)
    valData = makeProcessOrientated(valObs, P)
    t ← Root()
    GrowTree(trnData, t, F)
    PruneTree(valData, t)
    RevalueTree(trnData + valData, t)
    P+ = extractProcesses(t, P)
    extendModel(DM, P+)
end function
```

Figure 2: Pseudo code for the `Reprocess` approach, which refines a process description. The `Reprocess` function organises the data, grows the tree and replaces the original process with the generated refinements and supporting extensions (e.g., for maintaining active time features).

## Hypothesis Construction

The `Reprocess` method is presented in Figure 2 and presents the basic approach adopted in this work. The method uses both training and validation data sets, which are both preprocessed into process orientated data points as described in the previous section. For efficiency the features are evaluated in the state of each data point and stored. This also includes computing the associated effect modifying factor for each process effect.

At the heart of the approach (i.e., `GrowTree`) is a multi-target regression tree learning approach (De'Ath 2002) to learn a multi-target regression tree. The important difference that extends standard tree learning approaches is that the cost function sums error terms for each of the targets. The approach greedily identifies conditions that lead to the largest reduction in the error for the training data. This condition is used to split the data into two parts: those for which the condition holds and those for which it does not. In cases where the condition cannot be evaluated (e.g., a missing value) a third branch is created. A child is made for each of the data sets and the `GrowTree` process is repeated at each child with the associated (and therefore reduced) data set. The recursive process is stopped when there are too few data points at a node, or when splitting has a small change on the error.

As tree learning approaches can lead to over-fitting, we use a validation data set in order to inform a tree pruning process. Starting from the leaf nodes, the process re-evaluates each of the branchings of the tree and decides whether it provides a sufficient information gain, given the new data set. If it does not then the node becomes a leaf and the branch is pruned. The tree is completed by re-evaluating the values at the leaf nodes using the combined data sets.

**Feature Selection**   As the set of features generated to represent our machine learning problem are automatically and systematically generated, the first step is to identify a reduced subset of the features that appear most relevant for using while learning. We have considered two approaches in this work. The first is hand selection. The second is to exploit attribute selection, which is a common preprocessing step in machine learning applications. We have adopted a filter method based on correlation, which attempts to identify features that are correlated with the targets, yet uncorrelated with each other (Hall 1998).

**Describing Tree Node Conditions**   The system must also generate possible division points for the domains of numeric functions (e.g., $(< (/ ( occupancy ?l1) (capacity ?l1))\textbf{0.8})$). We adopt the standard approach for regression trees: calculate the value of a feature at each data point (at the current node), order the values and propose splitting the feature's domain in between each pair of adjacent values.

**The target**   The value of a tree leaf is calculated for each of the process's effects separately ($e \in EFF(P)$) as the average of the factors between the modelled value ($mod_d(e)$) and the observed values ($obs_d(e)$) for the data set ($DB$) at the leaf node:

$$t.e = [\textstyle\sum_{d \in DB} \frac{obs_d(e)}{mod_d(e)}]/|DB|$$

**Leaf Cost Function**   The evaluation of a hypothesis is the sum of the error at each of the tree's leaves and for each effect. This is based on the data at that leaf ($DB$) and calculates the squared differences between the observed values and the modified modelled value (using the tree's value) for each data point. In order to use a consistent measure of error we evaluate the error as the error of the updated process's effect:

$$Err = \textstyle\sum_{d \in DB} \sum_{e \in EFF(p)} [obs_d(e) - t.e * mod_d(e)]^2$$

**Extracting PDDL+ Processes From a Tree**   The final step is to modify the domain model with the refined process. The tree conditions themselves are expressible in PDDL. As a result the path to each leaf node describes a conjunction of conditions that should each hold for the leaf to be appropriate for the state. As such the tree is descended and conditions are recorded at each node (either $(<= X)$ or $(notX)$, for the left hand branch or $(> X)$ or $(X)$ for the right hand branch). Any extra branches for missing function values are identified and additional symbols are added to signify them in the initial state. These conditions are added to the original conditions to form a larger conjunction. The effects are then each modified by multiplying the right hand side of the effect terms by the associated factors learned for the leaf node. We then add a unique symbol to the process name to distinguish it from the processes constructed for the other leaves. These processes replace the existing process description.

The tree hypothesis is further examined to determine if it exploits any features that relate to the active time of the process. If the original model does not already represent the active time of the process (and the tree uses the features) then the model is further extended. In particular, for a process, $p$, we define a new set of functions (one for each instantiation of the process): $\langle p.\texttt{name}\rangle\_\texttt{counter}$. These functions are maintained by events that start and end the timer and a new process that records the active time.

## Evaluation

The aim of the evaluation is to determine whether the `Reprocess` procedure can improve simulation accuracy and lead to better plans. In this context we interpret a *better* plan as one that concludes in a state that is closer to the goal (during execution). We first describe the process of applying our approach to the UTM problem domain and then we present the results.

### Process Refinement in UTM

For this evaluation we have selected a portion of the road network for a major European city. A microscopic model was obtained from the transport authorities and has been captured in AIMSUN (Barceló and Casas 2005) professional modelling and simulation software. We have identified two networks at either side of the city for the experiments that we denote RHS and LHS. For each run, data (including turn-rates, active signals and occupancies) is collected from the simulator at regular intervals. For the experiments we have generated 8, 1 hour long, data sets for each of the networks. These data sets each start from different initial states and have been generated by first planning from that initial state using the original model and then simulating the original model's plan. For each network, 3 plans were further divided into training and validation sets. The remaining 5 simulations were used as the test set.

The starting (original) PDDL+ domain model is the representation of the UTM problem domain presented in (McCluskey and Vallati 2017), as introduced above. The turn-rates used to model flow in the original model are specified for each green time stage and link pair and approximate maximum average flow.

The UTM domain provides a challenging test case for this approach. AIMSUN uses a micro model of the road network, which has been heavily parameterised to represent individual PCUs (e.g., aggression level) and has been validated against real world data. This level of detail cannot be captured in a macro model; however, our expectation is that our base line model can be refined in order to better characterise the flow.

**The Refined Models**   We have learned two trees for each of the networks using the training and validation sets described above. Each uses a subset of the feature set generated by our system. The first (Hand) uses features that experts believed would determine in some way the real turn rate and the second (Auto) uses the automated process, described as follows.

There are 201 features generated (for max depth $\delta = 2$) from the original domain, which includes several that are not obviously meaningful (e.g., (- (occupancy ?r2) (max-greentime ?p))). For the Hand approach the following features were selected: density of in-link, density of out-link, proportion of process time (with respect to maximum stage-time) and approximated maximum turnrate.

For the Auto approach we used a filtering method for feature selection, which results in a subset of 8 (RHS) and 7

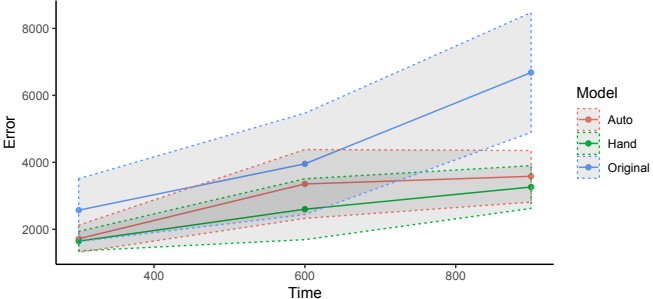

Figure 3: A comparison of the RHS network occupancies between modelled (Hand, Auto and Original) and those observed in AIMSUN after 300s, 600s and 900s of simulation.

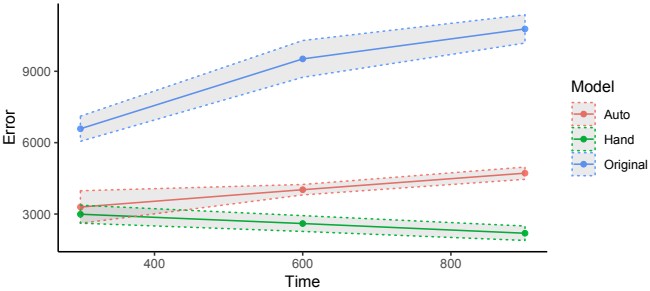

Figure 4: Comparing LHS network occupancies between modelled (Hand, Auto and Original) and those observed in AIMSUN. Dotted areas indicate standard deviation.

(LHS) features. The auto selected features include 2 (RHS) and 3 (LHS) of the 4 hand selected features.

### Accuracy Through Simulation

In this experiment we examine the accuracy of the model during plan simulation. During data collection in AIMSUN we have periodically recorded the occupancies of the links across the network. The plans that were used to generate the test data were simulated using the learned and original models. At each time point we compared the occupancies in the modelled states against the occupancies in AIMSUN.

We have plotted the squared error observed (between planning model and AIMSUN) for the Hand, Auto and Original models in the two networks (RHS in Figure 3 and LHS in Figure 4). The results indicate that the refined model using hand selected features leads to less error across the network in each of the networks. In the case of the automatic features, the refined model improves simulation accuracy in each of the subnetworks, although not as much as for the hand picked features. It is interesting to note that using automatically picked features for a different subnetwork does not result in effective learning.

**Efficiency of Refined Models**   The more refined the model, the more processes in the PDDL+ domain and consequently the bigger the simulation time. In Figure 5 we plot the simulation time (15 minutes simulating a plan on RHS

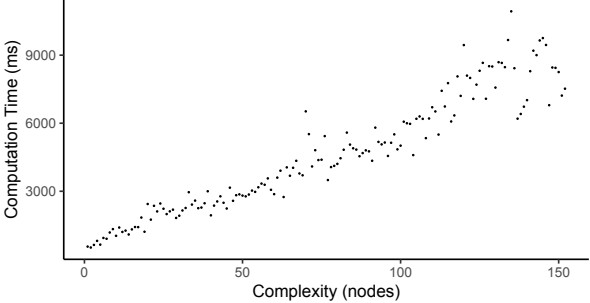

Figure 5: As the complexity of the refinement is increased, the computational effort of simulating the model increases.

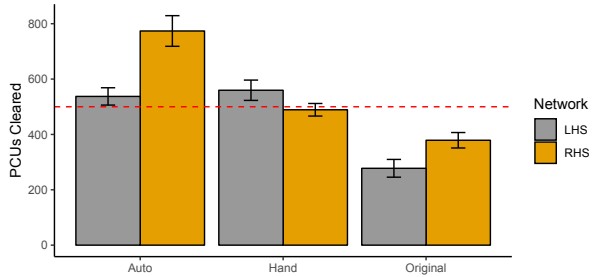

Figure 6: PCUs cleared from the goal link after simulating plans for the learned and original models to clear 500 PCUs (red line).

network) with increasingly complex models (from 0 to 150 decision tree nodes). It shows that computation time grows in a linear fashion as a function of the number of nodes in the tree. The trade-off between accuracy and computation time is application specific and will depend on the type of search that will be used.

### Accuracy in Planning

In this experiment we examine the accuracy of the models in the context of planning tasks in the UTM domain. In this case we ran AIMSUN three times from each initial state in our test set. In the first case we generated a plan using the original model and in the other cases, we generated a plan using the refined models. Each plan was simulated in AIM-SUN and stopped when the plan completed. At this stage the model had predicted that the goal would be achieved. We then analysed this final state in the simulator in order to determine whether the goal had been achieved.

For this experiment the goal was to clear 500 PCUs from a specific link. This was repeated 5 times in each network (LHS and RHS). We have plotted the number of PCUs that were cleared from the link at the end of the plan in the case of each model. Figure 6 shows that the original model over predicts the number of PCUs that are cleared from the goal link in each network. In comparison, Hand is able to more accurately predict the number of cleared PCUs in both networks and was particularly effective in the RHS network. The Auto model performs well in the LHS network (Fig-

ure 6). In the RHS the Auto model predicts that the clearance will be slower than was observed in AIMSUN. The selected features do not include the saturation of either the in- or out-links. It is therefore possible that the selected features did not allow an effective model to be learned, but instead a model that is over fitted to the plan distribution of the training data.

### Related Work

While much of the *motivation* of this work aligns with recent work on the challenges of domain model construction in Space applications, and in particular the ideas underlying the Interactive Model Development Environment (Clement et al. 2011), related work is centred around learning from execution data, an early example being Benson's TRAIL method, which used ILP on success and failure to learn action models in robot simulation environments (Benson 1996). Most related to this paper's refinement method is the progressive development of appropriate representations for concept learning, e.g., (Martin and Geffner 2000) and selection of the appropriate contexts for learning for control knowledge (Lindsay 2015) and heuristic correction (Yoon, Fern, and Givan 2007). In domin model acquisition (DMA) it has been common to assume accurate input data and this has allowed inductive learning approaches to be exploited, e.g., (Cresswell and Gregory 2011). In recent work, researchers have examined noisy data, exploiting clustering (Lindsay et al. 2017), machine learning (Zhuo and Kambhampati 2013) and deep learning (Asai and Fukunaga 2018) as part of their processes. DMA has progressively considered richer target fragments of the PDDL language, from propositional (Wu, Yang, and Jiang 2007; Cresswell and Gregory 2011), including ADL (Zhuo et al. 2010); to learning action costs (Gregory and Lindsay 2016) and numeric constraints (Segura-Muros, Pérez, and Fernández-Olivares 2018). As the richness of the language is increased, the space of possible models that explain the data vastly increases and has led to the DMA problem being set as either a search or learning problem (e.g., subtype selection in LOCM2; CP model to identify cost relevant parameters in NLOCM). We are not aware of any work in DMA that supports modelling of continuous transitions in PDDL+, although there are related works that consider numeric fragments, such as the approach in (Lanchas et al. 2007), which learns relational decision trees to appropriately estimate situation specific action durations from observational data.

### Conclusion and Future Work

In this paper we have presented an approach for refining hybrid planning models by exploiting observation data from executions. In order to exploit the information content of the original model, we discover how the effects of the original model can be modified in order to better fit the observed data. Our approach relies on learning a decision tree for each process, which captures the relationship between state functions and propositions and the effects of continuous processes. One advantage of our approach is that it can significantly reduce the knowledge engineering effort by generating its own refined PDDL+. We have presented both a

fully automated version, which selects its own features and a collaborative version, which takes advantage of expertly selected features. We have used our approach to refine a planning model for the Urban Traffic Management domain, which uses continuous processes to model the flow of traffic. We demonstrated that when hand selected features are used, the accuracy of simulation can be improved, resulting in a more accurate representation of occupancy of the network over time, as well as plans that are more effective during execution. We have shown that when using automatically selected features, it can also improve simulation accuracy, however, the selected features are not as effective. The current framework supports exploiting structure present in the original model, which is not always sufficient to capture important phenomena, e.g. a busy cross-flow turn can significantly alter turn rates if it ever gets full. In future work we will develop our approach within a framework for extending the planning model with additional features, e.g., based on derived predicates (de la Rosa and McIlraith 2011) or on identification of relevant features through structural analysis (Lindsay 2015). This will allow us to explore the space of possible features for continuous processes within a more general framework.

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
