# OpenReview forum: "Reprocess: Process Refinement for Improving Accuracy in Hybrid Planning Domain Models"
_icaps-conference.org/ICAPS/2019/Workshop/KEPS — KEPS 2019_

### Official Review · AnonReviewer1 · 2019-05-07
**Very interesting**

**Rating:** 5
**Confidence:** 3

**Review:**

This paper introduces an approach for refining PDDL+ process’ models description. The underlying idea is to generate plans using an existing —even if inaccurate— model, observe how the predicted execution differs from an actual (or simulated) execution, and refine the PDDL+ accordingly. For refining purposes decision trees are used, and are analysed for identifying conditions to be used for the different processes values. In fact, the refined domain models is likely to include a large number of processes, where each of them has different numerical effects, according to the surrounding conditions.

The paper is well written and easy to follow. The approach is well-motivated, and has the potential to foster the use of PDDL+ in real-world applications, as it reduces the pressure on knowledge engineers, and support maintainability.

As future areas of work, it would be great to assess whether different approaches for learning have the potential to overcome some of the issues of decision trees, or if more sophisticated features selection methods can substantially improve the accuracy of refined models.

---

### Official Review · AnonReviewer2 · 2019-05-09
**Learning more accurate numerical values in hybrid domain models**

**Rating:** 4
**Confidence:** 3

**Review:**

The paper addresses an important issue - accuracy of domain models. In particular, the paper utilizes Regression trees for adjusting values of numerical fluents such that they accurately capture real-world situation. The approach is evaluated in an Urban Traffic Control domain, where flows of vehicles have to be accurately represented so the planner can better reason about them.

I like the direction of the research as accuracy of domain models is an important factor determining usefulness and reliability of planning approaches, i.e., plans are  likely to be executable and better in terms of quality (when executed). The decrease of error is demonstrated in experimental evaluation.

I do not have much to criticize regarding paper content, its presentation and technical quality. Perhaps, the notion of the process could be formally introduced to make the paper more accessible for researchers outside the planning community. Also, a paper "Distributed Planning and Model Learning for Urban Traffic Control" by Pozanco, Fernandez and Borrajo, presented on KEPS 2018, should be mentioned as it has similar aims (combining learning and planning in Urban Traffic Control).